# The Antifungal Properties of *Tamarix aphylla* Extract against Some Plant Pathogenic Fungi

**DOI:** 10.3390/microorganisms11010127

**Published:** 2023-01-04

**Authors:** Fatimah Al-Otibi, Ghaida A. Moria, Raedah I. Alharbi, Mohamed T. Yassin, Abdulaziz A. Al-Askar

**Affiliations:** Botany and Microbiology Department, College of Science, King Saud University, Riyadh 11495, Saudi Arabia

**Keywords:** *Tamarix aphylla*, filamentous fungi, fungicidal properties, gas chromatography/mass spectrometry technique, Fourier-transform infrared spectroscopy, scanning and transmission electron microscopy, ethanolic and water extracts

## Abstract

*Tamarix aphylla* is a Saudi herb, which possesses antimicrobial properties and potentially introduces a solution to the subsequent dilemma caused by agrochemicals and antifungal misuse. The current study aimed to assess the fungicidal properties of water and ethanolic extracts of *T. aphylla* leaves against *Macrophomina phaseolina*, *Curvularia spicifera*, and *Fusarium* spp. The chemical composition of *T. aphylla* was evaluated by gas chromatography/mass spectrometry technique (GC–MS) and Fourier-transform infrared spectroscopy (FTIR). The antifungal assay assessed the fungal growth inhibition using the poisoned food technique. Scanning and transmission electron microscopy (SEM and TEM) were used to evaluate the structural changes induced in the fungal species post-treatment by *T. aphylla*. FTIR and GC–MS analysis revealed that *T. aphylla* extracts were rich in aromatic and volatile compounds, such as Benzeneselenol, Gibberellic acid, and Triaziquone, which proved multiple antifungal properties. The results showed significant inhibition in the growth of all species (*p* < 0.05) except for *F. moniliforme*, where the water extract induced the highest mycelial growth inhibition at the dose of 30%. The highest inhibition was for *M. phaseolina* treated with the water extract (36.25 ± 1.06 mm, *p* < 0.001) and *C. spicifera*, treated with the ethanolic extract (27.25 ± 1.77 mm, *p* < 0.05), as compared to the untreated control and the positive control of Ridomol. SEM and TEM revealed some ultrastructural changes within the fungal growth of treated *M. phaseolina*, which included the thickening and mild rupture of mycelia. Those findings suggested the robust antifungal properties of *T. aphylla* against some filamentous fungi. The phenolic composition illustrated the potential fungicidal properties of *T. aphylla*. Additional studies are required to focus on more antimicrobial properties of *T. aphylla* against other species, particularly those that might benefit the medical field.

## 1. Introduction

The high prevalence of fungal infections is an emerging health and socioeconomic dilemma for plants, humans, and animals [1]. This might be due to the ability of various fungi to infect almost all living organisms. Phytopathogenic fungi infections, in particular, are responsible for 20% of crop yield loss worldwide and 10% post-harvest [2], which threatens the growing demand for food supply. Studies showed that some species might be present in drinking water, so recently, they were added to the list of biological pollutants [3,4]. Momentarily, that dilemma was contained with the administration of antifungals.

The open distribution of systemic antifungals used in agribusiness met the growing demand for the agricultural stock, but over time, the misuse and overuse led to variable complications [5]. Firstly, phytopathogens mutated to mitigate the fungicide’s mode of action and/or grew resistant [6]. Secondly, studies showed antifungals alter the stages of plant growth, damaging crops and yielding economic loss [7,8]. Ridomil was involved in these formalities, which transgresses to secondary infections for consumers and might range from mild to systemic in immunocompromised patients [9].

One secondary pathway to human infection is through *fumonisins*, which threaten the corps culture. The mycotoxins of *Fumonisins* are produced by various *Fusarium* species, such as *F. solani*, *F. verticilliodes*, *F. proliferatum*, and *F. monliforme* [10,11,12]. *Fumonisins* are found in maize or maize-produced products [13]. Consumption of *F. verticilliodes* mycotoxin leads to health risks in vital organs (lungs, kidneys, skin), cancer, and others [14,15]. *F. solani* and *F. proliferatum* were isolated from date palms from various regions, one of which was Saudi Arabia [16]. Mycotoxin production was detected and reported at high rates of *F. proliferatum* in Saudi Arabia. Those species tested positive for various dangerous gene sequences [17].

*Macrophomina phaseolina* is a soil, seed, and stubble-born necrotrophic fungi [18]. It is responsible for more than 67 economic host–pathogen pairs, such as flowers, vegetables, and crops [18]. Wide hosts suggest nonhost-specific fungus, yet its infection is better known as charcoal rot. The name relates to the coloration resulting in the invasion and accumulation of black microsclerotia in host tissue [19]. The abundance of microsclerotia is a virulence factor responsible for secondary infection in immunocompromised patients [18].

*Curvularia spicifera* show extreme foliar pathogenicity in wheat and barley, most known as wheat crown rot disease, despite low virulent rates [20,21]. Isolates were identified from human eye infections [22]. *C. spicifera* is a keratinophilic fungus that transgresses into various mycoses in immunocompromised patients [23].

Phytomedicinal advancement became the optimal solution for microbial infections with the discovery or synthesis of natural medications for microbial infections. Owing to their durability, cost-effectiveness, positive sensitivity, and bioactivity, they were used as antimicrobial agents [24]. Bioactive compounds within botanic organisms are the synthetic basis of green chemistry as products of biopharmaceutical engineering. Bioengineered phytocompounds found in plants and herbs might manifest as repressors of microbial agents that cause primary or secondary infections in agriculture or mammalian mycotic infections [25].

A previous study suggested that Saudi Arabian extracts of the leaves possess valuable content of antioxidants, inflammatory, antibiotic, wound healing, astringent activity, antipyretic, and analgesic agents [26]. Secondary metabolites found include tannins, flavonoids, alkaloids, isoferulic acid, and ellagic acid, which contributed to the high antifungal rates in this plant [27].

One of these bioactive compounds was found in the native Saudi plant, *Tamarix aphylla* (*T. aphylla*). *T. aphylla* is a member of the Tamaricaceae family [27]. It grows mainly in the Middle East, Asia, and central and north Africa [27]. Many previous studies revealed the phytochemical composition of *T. aphylla*. They showed the rich content of phenylpropanoids, polyketides, alkaloids, and terpenoids, including gallic acid, ellagic acids, and tannins [28,29]. In folk medicine, *T. aphylla* has proven multiple medicinal uses for hypertension, abdominal pain, hair loss, cough, asthma, abscesses, wounds, rheumatism, jaundice, measles, aphrodisiac, diabetes, fever, and paralysis [29,30,31,32]. Furthermore, previous studies reported the antimicrobial properties of *T. aphylla* against *F. oxysporum*, *Aspergillus niger*, *A. fumigatus*, *A. flavus*, *Saccharomyces cerevisiae*, *Staphylococcus aureus*, *Bacillus subtilis*, *Escherichia coli*, *Salmonella typhi*, *Candida albicans*, and *Penicillium notatum* [27,33,34].

The current study aimed to evaluate the antifungal properties of *T. aphylla* obtained from the Saudi desert against five filamentous fungal species. These included *M. phaseolina*, *C. spicifera*, *F. verticilliodes*, *F. solani*, and *F. monoliforme*. To our knowledge, the fungicidal properties of *T. aphylla* have not been evaluated against these species, which highlight the novelty of this work.

## 2. Materials and Methods

### 2.1. Plant Material and Preparation of Extracts

Dried leaves of *T. aphylla* are commercially available. We purchased the herbal leaves from a local herbal market in Riyadh, KSA. The leaves were identified and taxonomic information was evaluated in the Department of Botany and Microbiology, College of Science, King Saud University, Saudi Arabia (Appendix A). Preparation of leaf extracts follows the instructions described by Alotibi and Rizwana (2019) [35]. Briefly, the leaves were washed by rinsing with distilled water to remove any adherent dust particles, then they were dried for 1–2 days at room temperature (25–30 °C), which was enough because of the dry weather of Riyadh city. Later, the dried leaves were finely ground to <10 µm particle diameter using a ESW-750 Lab Scale Horizontal Bead Mill (Shanghai ELE Mechanical and Electrical Equipment Co., Shanghai, China). The powder was weighed and kept at 4 °C until use.

In the current study, we prepared ethanolic and water extracts of the ground leaves of *T. aphylla*. In a separate container, 300 mL of each solvent was mixed with 30 g of prepared powder and homogenized on an orbital shaker (MaxQ 2000, Fisher Scientific, Vantaa, Finland) for two days. The macerates were filtered by centrifuge at 2000 rpm for 15 min, then reconstituted in their given concentration with their original solvents. The crude extracts were sterilized by filtration using a bacteriological sterile Sartolab^®^ RF vacuum filtration units of pore sizes 0.45 and 0.22 µm (Sartorius AG, Göttingen, Germany). Both extracts were redried to fine powders using a vacuum concentrator (Concentrator plus, Eppendorf, Hamburg, Germany) to allow the preparation of different concentrations for the following experiments. The dried products were mixed with Milli-Q water to prepare five different concentrations (0, 2.5, 7.5, 15, and 30%).

### 2.2. Microorganisms

*F. monoliforme*, *F. solani*, *F. verticilliodes*, *F. proliferatum*, *C. specificera*, and *Macrophomena phaseolina* were provided by the College of Food and Agricultural Sciences, Department of Plant Protection, King Saud University, Riyadh, Saudi Arabia. These species were cultured on potato dextrose agar (PDA) Petri dishes, as described before [35].

### 2.3. Antifungal Assay

The antifungal properties of *T. aphylla* crude extracts were evaluated by the Poisoned Food Technique, as described before [36]. Forty grams of PDA powder was mixed with 1 L of distilled water, boiled, mixed, and autoclaved for 15 min until homogenized. After semicooling, different concentrations of ethanolic or water extracts of *T. aphylla* (0, 2.5, 7.54, 15, and 30% in Milli-Q water) were mixed slowly with PDA and added to a sterile Petri dish (1 mL/plate). After solidification, 6 mm diameter pores were prepared in each culture plate using a sterile cork.

At the center of each pore, a 6 mm disc of each one-week fungal culture was added to each dish. The cultures were incubated for another week in 28 °C incubators. Other Petri dishes were used as positive controls using a known fungicide (Ridemol). Negative controls were made from PDA agar medium only, with Milli-Q water instead of the fungal species. The percentage of inhibition was calculated as follows:$$\%\;\mathrm{Mycelial}\;\mathrm{growth}\;\mathrm{Inhibition}=\frac{\mathrm{Cg}-\mathrm{Tg}}{\mathrm{Cg}}\times 100$$
where Cg indicates the mycelial growth induced by the negative control and Tg indicates the mycelial growth in the treated settings. All experiments were repeated in triplicates.

### 2.4. Scanning of the Fungal Growth by Scanning Electron Microscopy (SEM) and Transmission Electron Microscopy (TEM)

The fungal growth was tested at the minimum inhibitory concentrations by SEM and compared to the untreated control. Briefly, the treated and untreated fungi were suspended, centrifuged, and collected. The fungi were fixed on the microscope slides with 2.5% glutaraldehyde and kept overnight at 4 °C. On the next day, the slides were washed with phosphate buffer saline (1X) for 20 min and refixed with osmium tetroxide for 60 min. Later, the slides were dehydrated in serial dilutions of ethanol and dried by liquid CO_2_ in a critical-point dryer. Slides were coated with a thin layer of gold and visualized by the SEM microscope, JEOL JSM-6060LV (JEOL, Tokyo, Japan), with an accelerating voltage of 15 KV [35].

For TEM, the fungal samples were dehydrated. Samples showing the lowest MIC values were fixed and dehydrated in serial dilutions of ethanol, then embedded in resin. After solidification, the samples were sliced into 70–80 mm sections, as described for SEM. Post-dehydration, the test sample was embedded in resin and cut into fine sections (70–80 nm) using a Leica EM UC7 Ultramicrotome (Leica Microsystems, Wetzlar, Germany). These sections were treated with uranyl acetate and mounted on copper grids. The slides were visualized by a JEOL -1011 TEM microscope (JEOL, Tokyo, Japan) [35].

### 2.5. Fourier-Transform Infrared Spectroscopy (FTIR)

The chemical composition of the water extract of *T. aphylla* leaves was examined by FTIR to investigate the secondary metabolites that might participate in its fungicidal activities. The water extract was analyzed by a Nicolet-6700 FTIR system (Thermo Scientific, Waltham, MA, USA) in the range 400–4000 cm, according to the manufacturer’s instructions.

### 2.6. Gas Chromatography/Mass Spectrometry Technique (GC–MS)

The phenolic composition of the ethanolic extract of *T. aphylla* leaves was examined by GC–MS to investigate its content of flavonoids and other aromatic compounds that might explain its fungicidal activities. GC–MS analysis was performed by a thermo-gas chromatograph /mass spectrometer (Shimadzu, Kyoto, Japan). It was equipped with a 30 m long, 0.25 mm diameter, 0.25 μm film thickness Rtx-5MS capillary column (flow rate 1.2926 mL/min), a Triple-Axis Detector mass spectrometer (5975C V2-MSD), and G4513A-injector (Agilent Technologies, Santa Clara, CA, USA). The carrier used was helium gas at a maximum temperature of 280 °C. The chemical composition was evaluated by the commercial libraries of NIST14 https://www.nist.gov/ (accessed on 10 of May 2022) and Wiley https://sciencesolutions.wiley.com/solutions/technique/gc-ms/ (accessed on 10 of May 2022).

### 2.7. Statistical Analysis

The statistical significance was assessed by SPSS version 22 (IBM Corp., Armonk, NY, USA). The results were expressed as means ± standard deviation. The significance of inhibition was tested by chi-square (χ^2^) and the values were considered significant at *p* < 0.05.

## 3. Results

### 3.1. In Vitro Antifungal Activity

Antifungal activities of *T. aphylla* leaf extracts against different fungal species are shown in Figure 1. As indicated, all treatments had an obvious inhibition of the mycelial growth of all species. The water and ethanolic extracts at 15% did not induce any obvious inhibition for *F. proliferatum* and *F. moniliforme*, respectively.

Filamentous fungi showed significant inhibition by ethanol and water extracts. The water extract of *T. aphylla* leaves showed significant inhibition in the mycelial growth of *F. verticilliodes* (37.25 ± 3.54%), *M. phaseolina* (36.25 ± 1.06%), *C. spicifera* (36.25 ± 2.12%), *F. solani* (42 ± 1.77%), and *F. proliferatum* (61 ± 4.95%) at the dose of 30 mg/mL compared to the control (*p* < 0.05) (Figure 2, Table 1). Similarly, ethanolic extract at 30 mg/mL exhibited maximum mycelial growth inhibition of *F. verticilliodes* (30 ± 0.01%), *M. phaseolina* (23.75 ± 0.35%), *C. spicifera* (27.25 ± 1.77%), *F. solani* (37.25 ± 0.35%), and *F. proliferatum* (38.25 ± 3.18%) (*p* < 0.05) (Figure 3, Table 2). Thus, the results showed that the ethanolic extract of *T. aphylla* leaves had more inhibitory effects on the mycelial growth of the abovementioned fungi species than the water extract and even more than the positive control (Ridomel) (Table 3, Figure 4).

### 3.2. FTIR Analysis of the Water Extract of T. aphylla

Water extract of *T. aphylla* was analyzed through FTIR analysis to examine secondary metabolites responsible for antifungal activity. Results concerning *T. aphylla* showed the presence of volatile components. The infrared spectra provided relevant important information. The wide peaks at 3425 cm^−1^ and 2932 cm^−1^ resulted from the expansion of a single bond O-H. The peak at 1622 cm^−1^ indicated the expansion of the C=C bond of α, β-unsaturated ketone, while the peak at 1403 cm^−1^ indicated O-H bending of the carboxylic acid (Figure 5, Table 4).

### 3.3. Gas Chromatography–Mass Spectrometry Technique (GC–MS)

Phenolic constituents of the ethanolic extract, the most active phytochemicals of *T. aphylla*, were examined and identified through GC–MS analysis. The chemical structure of the resulting compounds was drawn by the free online software MolView https://molview.org/ (accessed on 15 May 2022). Out of eight compounds, the major components were the following: Benzeneselenol; Thiourea, 1-napthalenyl; Gibberellic acid; Naphthalene, 1-(1-methyl ethyl)-; 1, 3-Bendesenediamine, 2,4-dinitro-N3, N3-dipropyl-6-(trifluoromethyl); Triaziquone; and Retinol, 9-cis (Figure 6, Table 5).

### 3.4. SEM and TEM Scanning of M. phaseolina

In the current study, the fungal growth of *M. phaseolina* was scanned by SEM and TEM techniques in response to plant extract effectiveness. The species was chosen as it had the most robust inhibition of mycelial growth among other species. SEM images (Figure 7) show the untreated biomass (control) with natural mycelial/conidal growth a distinguishing morphological characterization, as represented in the left panel of the SEM micrographs. The right panel of the SEM micrographs showed the microscopic morphology of treated-fungal species growth. Both water and ethanolic extracts of *T. aphylla* induced the rapture and thickening of mycelia by 1.78 and 1.21 µm, respectively, as compared to 0.6 µm in the untreated control. Additionally, the color of the mycelia turned from gray in control to off-white puffy in the treated samples, which indicated ultrastructural modifications induced by the *T. aphylla* extracts (Figure 7).

Similarly, TEM showed ultrastructural changes in the fungal growth of M. phaseolina in response to treatments by *T. aphylla* extracts. The untreated biomass (control) of M. phaseolina had natural mycelial and conidial structures with distinguishing internal characterizations as represented in the left panel of the TEM micrographs. The right panel shows that both treatments caused the accumulation of large white vacuoles. Those vacuoles caused the rapture of the cellular wall, which might illustrate the inhibitory effects of the mycelial growth of M. phaseolina by *T. aphylla* extracts (Figure 8).

## 4. Discussion

*T. aphylla* is a wild edible plant; not expensive and significantly contributes to human health improvement in terms of the cure and prevention of diseases. Therefore, their extracts serve and are suitable to be nontoxic to mammals. Their high efficacy for controlling plant pathogens would make them a good alternative in integrated plant protection, which might reduce the human health hazard associated with certain synthetic fungicides [26,27].

In the current study, water and ethanolic extracts of *T. aphylla* inhibited the growth of *F. verticilliodes*, *M. phaseolina*, *C. spicifera*, *F. solani*, and *F. proliferatum* species at different concentrations. Furthermore, TEM and SEM scanning showed ultrastructural modifications in the mycelial and conidial structures of the treated *M. phaseolina*. These findings agreed with previous studies showing that the active components found in *T. aphylla* extract disrupted and altered fungal growth [27,37]. The inhibition rate of *M. phaseolina* by the water extract of *T. aphylla* leaves was the greatest at 36.25 ± 1.06 mm, which induced almost 60% inhibition, compared to 90 ± 0.01 mm in the untreated control. The microscopic images showed altered cellular damage, elongated hyphae, poor stain absorption, and disruption in spore formation. The color changed from gray mycelia to off-white puffy and elevated mycelia. In the study by Alrumman (2016), water-extracted *T. aphylla* leaves affected the microbial growth of *S. aureus*, *K. pneumoniae*, *K. oxytoca*, *P. mirabilis*, *P. aeruginosa*, *M. luteus*, and *Shigella* sp., and one pathogenic *Candida* sp. [37]. Another study showed that different extracts of *T. aphylla* at 0.078–2.5 mg/mL dose induced significant antimicrobial effects against *S. aureus*, *Enterococcus faecalis*, *E. coli*, *P. aeruginosa*, and *Acinetobacter baumannii* bacteria [38].

While certain compounds were not detected in previous research, the aerial part of *T. aphylla* is known to be rich in the most prevalent compounds, such as polyphenols and flavonoids, providing them with the greatest antioxidant activity [25]. Polyphenols are secondary metabolites that assist in the response of oxidative stress generated by reactive oxygen species (ROS). Under abiotic conditions, *T. aphylla* increased that phenolic content. Alkaloids are known for altering membrane permeability, altering fungal efflux pumps, and targeting cell wall pathways. Further results include instability and disintegration leading to cellular death [36].

In the current study, water and ethanolic extracts of *T. aphylla* inhibited the growth of F. *verticilliodes*, *M. phaseolina*, *C. spicifera*, *F. solani*, and *F. proliferatum* species at different concentrations. Furthermore, TEM and SEM scanning showed ultrastructural modifications in the mycelial and conidial structures of the treated *M. phaseolina*. These findings agreed with previous studies showing that the active components found in *T. aphylla* extract disrupted and altered fungal growth [27,37]. *M. phaseolina’s* inhibition rate by the water extract of *T. aphylla* leaves was the greatest at 36.25 ± 1.06 mm, which induced almost 60% inhibition, compared to 90 ± 0.01 mm in the untreated control. The microscopic images showed altered cellular damage, elongated hyphae, poor stain absorption, and disruption in spore formation. The color changed from gray mycelia to off-white puffy and elevated mycelia.

The present study evaluated the antimicrobial activity of the leaves of a Saudi Arabian weed, *T. aphylla*, against a range of phytopathogenic fungi strains. The results showed that the water and ethanolic extracts of *T. aphylla* leaves possessed significant antimicrobial activities against the phytopathogens of *C. spicifera*, *M. phaseolina*, and *F. verticillioides.* Recently, the increased resistance of some pathogens to antibiotics due to uncontrolled excessive usage encourages scientists and physicians to devote more effort toward investigating other substitutions. That included chemotherapeutic drugs and natural bioactive materials as they are inexpensive and with limited or no side effects [38].

In the current study, the GC–MS analysis revealed that *T. aphylla* leaves are rich in Benzeneselenol (42.35%), 9-cis Retinol (22.28%), Triaziquone (14.01%), and Gibberellic acid (11.21%). A previous study showed that some derivatives of Benzeneselenol possessed significant antifungal activities in some *Candida* and *Aspergillus* species [39]. A recent study showed that trans Retinoic acid at 0.5 and 1 mM had significant fungicidal activity against *Aspergillus fumigatus* [40].

Antimicrobial activities of *T. aphylla* (L.) were investigated to explore its medicinal importance. Preliminary screening by secondary metabolites revealed the presence of flavonoids, sterols, terpenoids, alkaloids, and tannins in a methanolic extract of the stem bark of *T. aphylla* (L.) [41]. The presence of secondary metabolites is responsible for its different pharmacological activities [42]. A previous study showed that the flavonoid antifungal activity is probably due to its ability to complex with extracellular and soluble proteins and cell walls, as described for quinines, which are more lipophilic flavonoids that may disrupt microbial membranes [43]. Moreover, the mechanisms of phenol toxicity against microorganisms include enzyme inhibition by the oxidized compounds, possibly through reaction with sulfhydryl groups or through nonspecific interactions with the proteins, as shown by Biradar et al. (2008) [44].

The phenolic phytochemicals and organic acids had an antimicrobial effect. That is due to their ability to induce hyperacidification via proton donation at the plasma membrane interface of the microorganism and intracellular cytosolic acidification. That may disrupt the H+-ATPase required for ATP synthesis [45,46]. Initially, the hydrophobic biphenolic compounds (rosmarinic acid and ellagic acid) are likely to bind on the plasma membrane, cell wall, and lipopolysaccharide–water interface of the cell without penetration [47]. These phenolic phytochemicals might stack on the plasma membrane, which affects the membrane fluidity and destabilization and results in partial disruption. That allows some phenolics, such as hydroxyl benzoic acid, chlorogenic acid, gallic acid, and lactate, to enter the cytosol [48].

Numerous factors can affect the biological activity of certain phytochemical compounds when in contact with microbial cells (pathogen) and plant tissue (host). Conceivably, in the complex host/antimicrobial compound/pathogen system, many biochemical processes can occur with different effects on the biological activity of the antimicrobial compound. It might be hypothesized that the molecules of the applied antimicrobial compounds can undergo ultrastructural changes (degradation, hydrolysis, polymerization, etc.), which causes an increase or loss in their original biological activity. Furthermore, the substances in the extracts may act as elicitors of resistance through different mechanisms mediated by the host tissue [49,50].

## 5. Conclusions

In conclusion, the present study highlights the antifungal activities of the Saudi native-grown plant, *T. aphylla.* The study reveals that the water and ethanolic extracts of *T. aphylla* might act as antifungal agents against some relevant phytopathogenic fungi, such as *F. verticilliodes*, *F. solani*, *F. monliforme*, *M. phaseolina*, and *C. spicifera*. The phytochemical composition of *T. aphylla* is rich in secondary metabolites, flavonoids, alkaloids, and aromatic compounds. Those compounds exhibit several phytomedicinal and antimicrobial properties, which allow them to be studied as possible medicinal alternatives. Fungal standard deviation was recorded, where the morphology of fungal adaptation to extracts showed some ultrastructure modifications. Further studies need to be performed to classify the range of fungal sensitivity in separate extract concentrations.

## Figures and Tables

**Figure 1 microorganisms-11-00127-f001:**
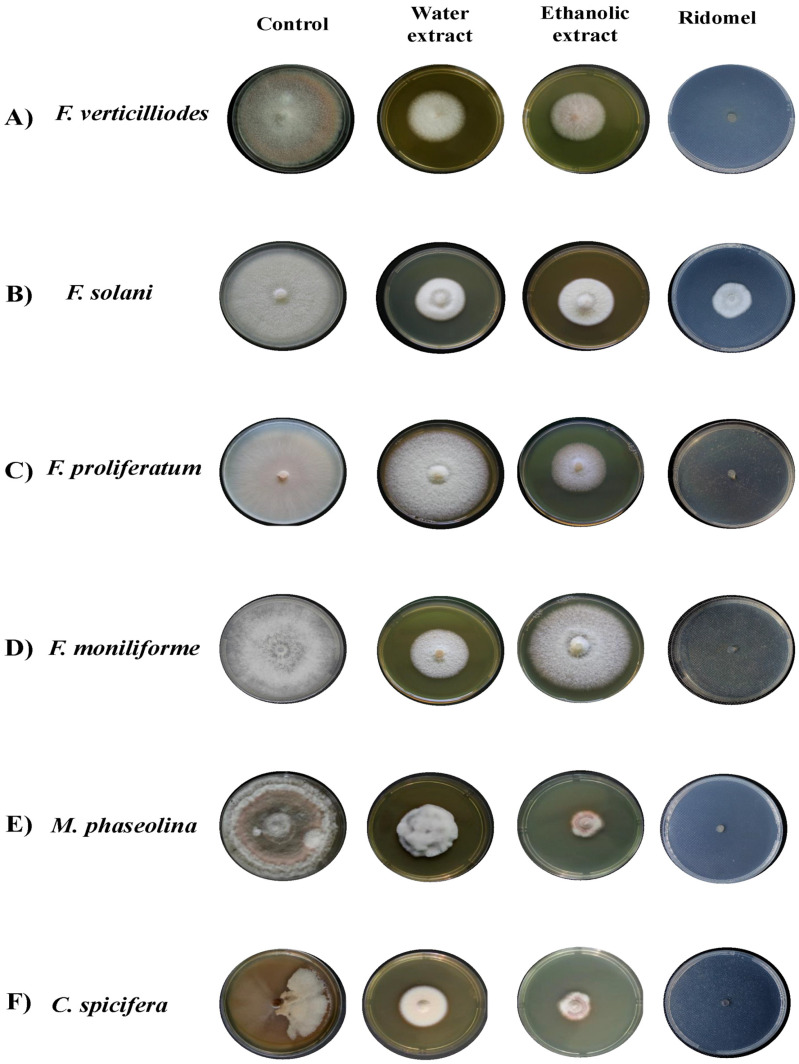
Antifungal effects of *T. aphylla* leaf extracts against different phytopathogenic fungi. The species were growing on PDA Petri dishes and treated with either ethanolic or water extracts of *T. aphylla* (30%). The mycelial growth was compared to the untreated control and the positive fungicide (Ridomil, 0.5 mg/mL). (**A**) *F. verticilliodes*, (**B**) *F. solani*, (**C**) *F. proliferatum*, (**D**) *F. moniliforme*, (**E**) *M. phaseolina*, and (**F**) *C. spicifera*.

**Figure 2 microorganisms-11-00127-f002:**
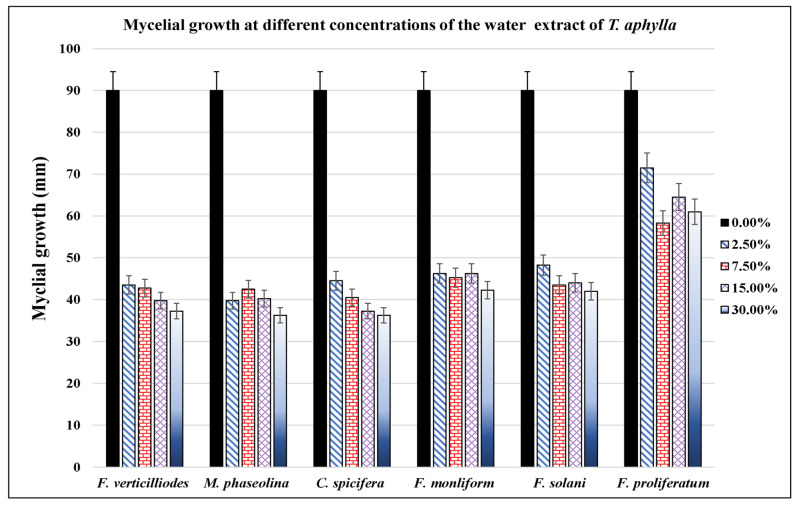
Effect of different concentrations (0, 2.5, 7.5, 15, and 30%) of *T. aphylla* water extract on growth of different phytopathogenic fungi. The extract showed variable inhibitory effects on the growth of different phytopathogenic fungi cultured on PDA agar medium. Bars indicate the standard error.

**Figure 3 microorganisms-11-00127-f003:**
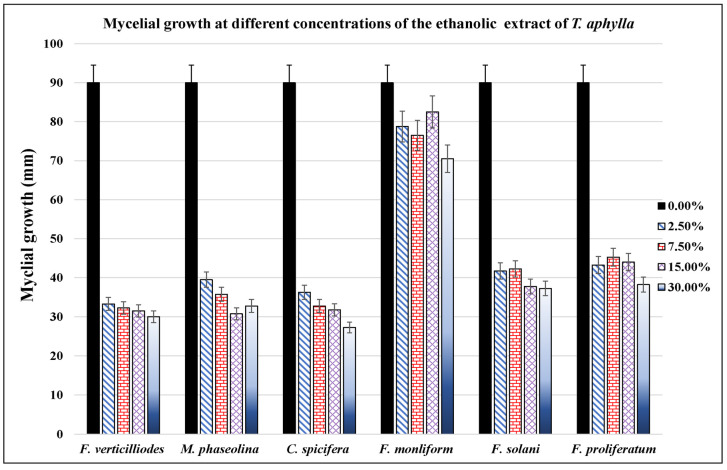
Effect of different concentrations (0, 2.5, 7.5, 15, and 30%) of *T. aphylla* ethanolic extract on growth of different phytopathogenic fungi. The extract showed variable inhibitory effects on the growth of different phytopathogenic fungi cultured on PDA agar medium. Bars indicate the standard error.

**Figure 4 microorganisms-11-00127-f004:**
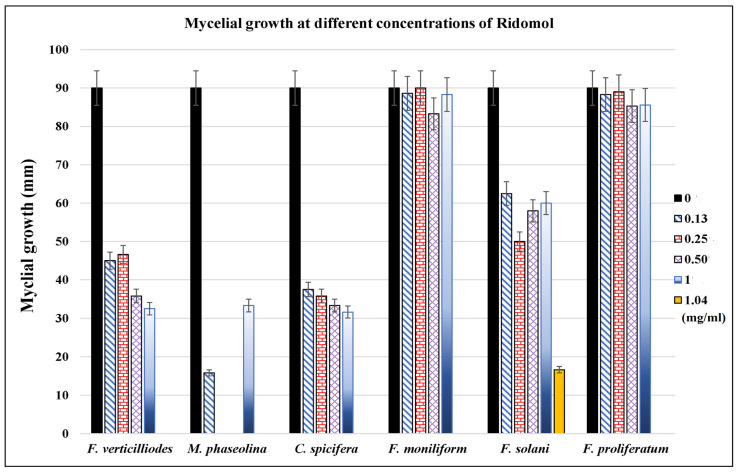
Effect of different concentrations (0, 0.13, 0.25, 0.5, 1, and 1.4 mg/mL) of Ridomol on growth of different phytopathogenic fungi. The antifungal agent showed variable inhibitory effects on the growth of different phytopathogenic fungi cultured on PDA agar medium. Bars indicate the standard error.

**Figure 5 microorganisms-11-00127-f005:**
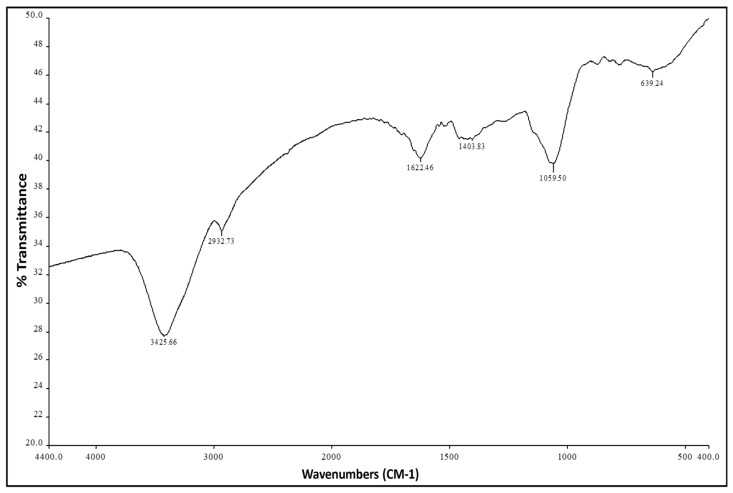
FTIR assay analysis of the water extract of *T. aphylla.* The results were produced by a Nicolet 6700 FT-IR Spectrometer in the range 500–4000/cm.

**Figure 6 microorganisms-11-00127-f006:**
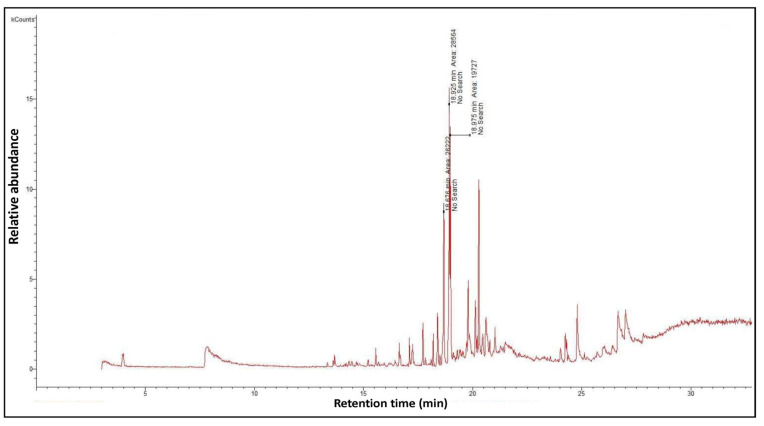
Gas chromatography–mass spectrometry (GC–MS) analysis of *T. aphylla* ethanolic extract.

**Figure 7 microorganisms-11-00127-f007:**
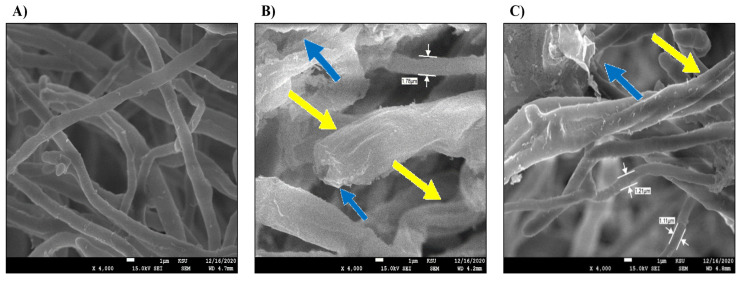
SEM images of the fungal growth of *M. phaseolina*. The treated fungus had swelling (yellow arrows) and rapture (blue arrows) of conidial surface compared to the control: (**A**) untreated control, (**B**) effect of the water extract of *T. aphylla*, (**C**) effect of the ethanolic extract of *T. aphylla*.

**Figure 8 microorganisms-11-00127-f008:**
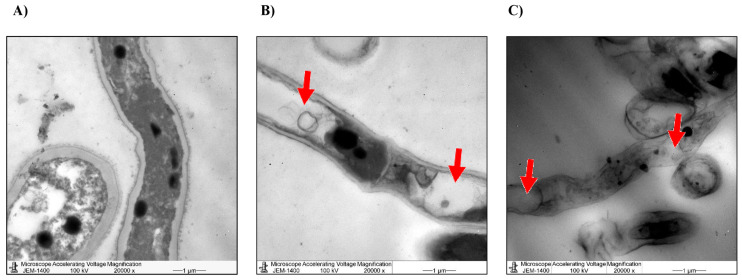
TEM images of the fungal growth of *M. phaseolina.* The treated fungi showed the accumulation of large white vacuoles (red arrows), which caused the rapture of the cellular wall, compared to the control: (**A**) untreated control, (**B**) effect of the water extract of *T. aphylla*, (**C**) effect of the ethanolic extract of *T. aphylla*.

**Table 1 microorganisms-11-00127-t001:** Effect of different concentrations of *T. aphylla* water extract on growth of different phytopathogenic fungi.

Doses	*F. verticilliodes*	*M. phaseolina*	*C. spicifera*	*F. moniliforme*	*F. solani*	*F. proliferatum*
0%	90 ± 0.01	90 ± 0.01	90 ± 0.01	90 ± 0.01	90 ± 0.01	90 ± 0.01
2.5%	43.5 ± 3.54	39.75 ± 1.06	44.5 ± 2.12	46.25 ± 1.06	48.25 ± 1.77	71.5 ± 4.95
7.5%	42.75 ± 3.89	42.5 ± 1.41	40.5 ± 0.71	45.25 ± 1.06	43.5 ± 1.41	58.38 ± 19.98
15%	39.75 ± 0.35	40.25 ± 0.35	37.25 ± 1.77	46.25 ± 1.06	44 ± 0.71	77.25 ± 18.03
30%	37.25 ± 3.54	36.25 ± 1.06	36.25 ± 2.12	42.25 ± 1.06	42 ± 1.77	61 ± 4.95
*p*-value	<0.001 *	<0.001 *	<0.001 *	0.053	0.001 *	0.006 *

* Significant *p*-value (<0.05).

**Table 2 microorganisms-11-00127-t002:** Effect of different concentrations of *T. aphylla* ethanolic extract on growth of different phytopathogenic fungi.

Doses	*F. verticilliodes*	*M. phaseolina*	*C. spicifera*	*F. moniliforme*	*F. solani*	*F. proliferatum*
0%	90 ± 0.01	90 ± 0.01	90 ± 0.01	90 ± 0.01	90 ± 0.01	90 ± 0.01
2.5%	32.25 ± 0.35	39.5 ± 0.71	36.25 ± 1.77	78.75 ± 0.35	41.75 ± 1.06	43.25 ± 1.06
7.5%	32.25 ± 0.35	35.75 ± 2.47	32.75 ± 0.35	76.5 ± 0.71	42.25 ± 2.48	45.25 ± 1.77
15%	31.5 ± 0.71	30.75 ± 3.18	31.75 ± 2.47	82.5 ± 2.83	37.75 ± 1.06	44 ± 0.01
30%	30 ± 0.01	32.75 ± 0.35	27.25 ± 1.77	70.5 ± 2.83	37.25 ± 0.35	38.25 ± 3.18
*p*-value	0.002 *	0.015 *	0.011 *	0.163	0.001 *	0.013 *

* Significant *p*-value (<0.05).

**Table 3 microorganisms-11-00127-t003:** Effect of different concentrations of Ridomol on growth of different phytopathogenic fungi.

Doses	*F. verticilliodes*	*M. phaseolina*	*C. spicifera*	*F. moniliforme*	*F. solani*	*F. proliferatum*
0 mg/mL	90 ± 0.01	90 ± 0.01	90 ± 0.01	90 ± 0.01	90 ± 0.01	90 ± 0.01
0.125 mg/mL	45 ± 0.01	15.83 ± 27.42	37.5 ± 0.01	88.67 ± 1.53	62.5 ± 12.99	88.33 ± 2.89
0.25 mg/mL	46.67 ± 2.89	0	35.83 ± 1.44	90 ± 0.01	50 ± 17.32	89 ± 1.73
0.5 mg/mL	35.83 ± 1.44	0	33.33 ± 2.89	83.33 ± 1.53	58.33 ± 2.89	85.33 ± 1.53
1 mg/mL	32.5 ± 0.01	50 ± 7.07	31.67 ± 3.82	88.33 ± 2.89	60 ± 0.01	85.67 ± 1.53
*p*-value	<0.001 *	0.012 *	<0.001 *	0.146	0.005 *	0.003 *

* Significant *p*-value (<0.05).

**Table 4 microorganisms-11-00127-t004:** FTIR analysis of the water extract of *T. aphylla*.

Absorption (cm^−1^)	Group	Compound Class
3425	O-H stretching	alcohol
2932	O-H stretching	alcohol
1622	C=C stretching	α, β-unsaturated ketone
1403	O-H bending	carboxylic acid
1266	C-N stretching	aromatic amine
639	C-Br stretching	halo compound

**Table 5 microorganisms-11-00127-t005:** Qualitative phytochemical analysis showing the phenolic constituents of the ethanolic extract of *T. aphylla*.

Phenolic Compound	Structure	%	Molecular Weight (M.W.)	Chemical Formula
Benzeneselenol	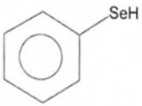	42.35	157.07 g/mol	C6H6Se
Thiourea, 1-napthalenyl-	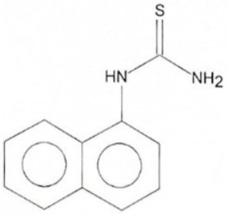	6.23	202.28 g/mol	C11H10N2S
Gibberellic acid	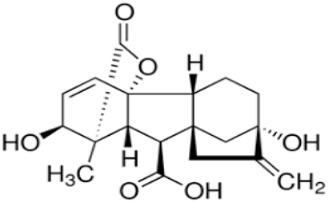	11.21	346 g/mol	C19H22O6
Naphthalene, 1-(1-methylethyl)-	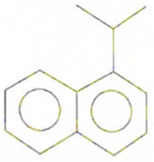	1.39	170.25 g/mol	C13H14
1, 3-Bendesenediamine, 2,4-dinitro-N3, N3-dipropyl-6-(trifluoromethyl)-	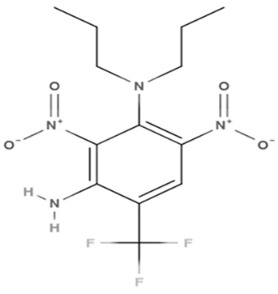	2.53	350 g/mol	C13H17F3N4O4
Triaziquone	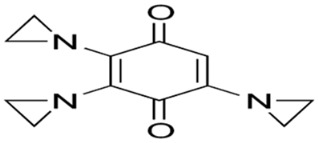	14.01	231 g/mol	C12H13N3O2
Retinol, 9-cis	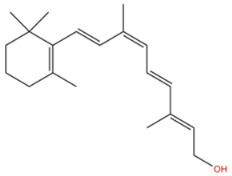	22.28	284 g/mol	C20H30O

## Data Availability

All the data presented in this study are available within the current article. All statistical analysis results and raw data are available on request from the corresponding author.

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
