# Peer review of "The Antifungal Properties of Tamarix aphylla Extract against Some Plant Pathogenic Fungi"

_microorganisms, 2023, doi:10.3390/microorganisms11010127_

Round 1

Reviewer 1 Report

·         Remove the dot at Line 3 of title

·         The abstract should be rephrased to clarify the results in easier way

·         The Ms should be checked by expert in English language to better describe the text and correct several grammatical errors

Rephrase the sentence (the results were obscure ………etc  L13-14) to be clear

·         The authors should provide more details about the used plant Tamarix aphylla with photo, its abundance, amount, problems caused (if any) , uses , economics and conditions.

·         -correct CO2 at L 124

·         At figure 1, please show the microbial names on the figures for better explanations and show the used concentration of Ridomil  

·         Figure 2 and figure 4 are exactly same??? Please provide the correct figures

·         At table 5, please show the amount or ratio of ach compound? What is the most prevalent and the major compound that show the effects should be discussed?

·         Table 5 correct the formatting of chemical formula

·         Correct the reference to be in the journal format

Author Response

Comments and response to reviewer No. 1

  • Remove the dot at Line 3 of title.

Response: We agree with the reviewer’s comment. We have corrected the title.

  • The abstract should be rephrased to clarify the results in easier way.

Response: We agree with the reviewer’s comment. We have upgraded the abstract as required.

  • The Ms should be checked by expert in English language to better describe the text and correct several grammatical errors.

Response: We agree with the reviewer’s comment. We have made excessive lingual, grammatical, and punctuational editing all over the manuscript. 

  • Rephrase the sentence (the results were obscure ………etc, L13-14) to be clear.

Response: We agree with the reviewer’s comment. We have upgraded the abstract as required.

  • The authors should provide more details about the used plant Tamarix aphylla with photo, its abundance, amount, problems caused (if any), uses, economics and conditions.

Response: We agree with the reviewer’s comment. We have updated the introduction section with the required information. The photo of the plant was included in the materials and methods section, as Supplementary Figure S1.

  • correct CO2 at L 124.

Response: We agree with the reviewer’s comment. We have corrected the required information.

  • At figure 1, please show the microbial names on the figures for better explanations and show the used concentration of Ridomil.

Response: We agree with the reviewer’s comment. We agree with the reviewer’s comment. We have updated Figure 1 and figure 1 caption as required.

  • Figure 2 and figure 4 are exactly same??? Please provide the correct figures.

Response: We agree with the reviewer’s comment. We have updated Figures 2 and 4 as required.

  • At table 5, please show the amount or ratio of ach compound? What is the most prevalent and the major compound that show the effects should be discussed?

Response: We agree with the reviewer’s comment. We have updated Table 5 as required.

  • Table 5 correct the formatting of chemical formula.

Response: We agree with the reviewer’s comment. We have updated Table 5 as required.

  • Correct the reference to be in the journal format.

Response: We agree with the reviewer’s comment. We have updated the references section as required.

Reviewer 2 Report

The studies on antifungal properties of plants is becoming a major challenge in the face of food and environmental contamination. Thus, the main goal of this research to evaluate antifungal properties of Tamarix aphylla against Fusarium spp., Curvularia spicifera and Macrofomina phaseolina is relevant in both scientific and practical terms. It is extremely important that fumonisins are classified as possibly carcinogenic in human and animals. However, research and its presentation could be characterized as quite limited.

The title and keywords of the article match the content reasonably well.

Abstract is adequate and contains short information on results of this study.

Introduction. In the introduction, the authors examine the problems caused by dangerous fungal diseases that affect the cultivation of various plants and release chemical compounds noxiuos to human or animal health. Thus, the relevance is justified, but the novelty of the planned research is not discussed. Maybe it will be new or adapted methods, identified chemical compounds, etc. This should be covered in this section. What did the authors mean in the sentence (p. 2, line 65) when they write "botanic plants and herbs". What are "botanic plants" and why are herbs not plants?

Materials and methods. Data processing and methods used are presented in this section.

Results. The obtained data are presented in 5 tables and 8 figures. The results are described in a rather fragmentary manner, which makes it difficult to understand their essentiality. For example, Fig. 1 shows the results of the antifungal effect with both ethanolic and aqueous 0.5% extracts of T. aphylla. Later, the study was carried out with 4 concentrations of extracts. In Fig. 2, it is indicated that the average inhibition zones induced by Ridomel are presented, but the title indicates "antifungal properties of effect of T. aphylla aqueous extract”. Next, what is the difference between Fig. 2 and Fig. 4? The authors should justify why the morphological characteristics are given only for Macrophomina phaseolina. P. 11, lines 236-238, it is necessary to indicate which publications are meant. I would suggest to the authors to avoid, as far as possible, the repetitive information in the text that is given in the tables. Therefore, I would suggest that authors should carefully review the tables and figures and description of results.

Discussion. The discussion is the place in the manuscript where authors should discuss their research findings in the context of other studies. However, much of this chapter is more narrative in nature. I would suggest a more careful review and specification of p. 13, lines 287-312.

Conclusions. The conclusions briefly discuss the aim and research methods, but I missed the generalization and the significance of this research. I would suggest abandoning the discussion of the methods used (p. 13, lines 319-324), presenting the substantive results and supporting the state-of-the art of this study.

There are many typographical mistakes in the article. I indicate a few of them: p. 1, line 22; p. 2, line 55; p.3, line 111.

Author Response

  • The title and keywords of the article match the content reasonably well.

Response: We agree with the reviewer’s comment. Thanks a lot.

  • Abstract is adequate and contains short information on results of this study.

Response: We agree with the reviewer’s comment. We have upgraded the abstract as required.

  • In the introduction, the authors examine the problems caused by dangerous fungal diseases that affect the cultivation of various plants and release chemical compounds noxiuos to human or animal health. Thus, the relevance is justified, but the novelty of the planned research is not discussed. Maybe it will be new or adapted methods, identified chemical compounds, etc. This should be covered in this section. What did the authors mean in the sentence (p. 2, line 65) when they write "botanic plants and herbs". What are "botanic plants" and why are herbs not plants?

Response: We agree with the reviewer’s comment. We have updated the introduction section with the required information.

  • Materials and methods. Data processing and methods used are presented in this section.

Response: We agree with the reviewer’s comment. Thanks a lot. Some updated were added to clarify some points regarding the microorganisms and preparation of extracts.

  • The obtained data are presented in 5 tables and 8 figures. The results are described in a rather fragmentary manner, which makes it difficult to understand their essentiality. For example, Fig. 1 shows the results of the antifungal effect with both ethanolic and aqueous 0.5% extracts of T. aphylla. Later, the study was carried out with 4 concentrations of extracts. In Fig. 2, it is indicated that the average inhibition zones induced by Ridomel are presented, but the title indicates "antifungal properties of effect of T. aphylla aqueous extract”. Next, what is the difference between Fig. 2 and Fig. 4? The authors should justify why the morphological characteristics are given only for Macrophomina phaseolina. P. 11, lines 236-238, it is necessary to indicate which publications are meant. I would suggest to the authors to avoid, as far as possible, the repetitive information in the text that is given in the tables. Therefore, I would suggest that authors should carefully review the tables and figures and description of results.

Response: We agree with the reviewer’s comment. Some mistakes were obtained from the online submission of the figures, we have updated the results section accordingly. For the comment in P. 11, lines 236-238, we removed that part as we feel, it’s not necessary.

  • The discussion is the place in the manuscript where authors should discuss their research findings in the context of other studies. However, much of this chapter is more narrative in nature. I would suggest a more careful review and specification of p. 13, lines 287-312.

Response: We agree with the reviewer’s comment. We have updated the discussion and conclusion part as required.

  • The conclusions briefly discuss the aim and research methods, but I missed the generalization and the significance of this research. I would suggest abandoning the discussion of the methods used (p. 13, lines 319-324), presenting the substantive results and supporting the state-of-the art of this study.

Response: We agree with the reviewer’s comment. We have updated the discussion and conclusion part as required.

  • There are many typographical mistakes in the article. I indicate a few of them: p. 1, line 22; p. 2, line 55; p.3, line 111.........

Response: We agree with the reviewer’s comment. We have made excessive lingual, grammatical, and punctuational editing all over the manuscript. 

Reviewer 3 Report

The authors didn't show any novelty in their research. This research idea and design did many times on Tamarix aphylla extract against plant pathogenic fungi.

Author Response

  • The authors didn't show any novelty in their research. This research idea and design did many times on Tamarix aphylla extract against plant pathogenic fungi.

Response: We agree with the reviewer’s comment. We have updated the introduction section with the required information.

Reviewer 4 Report

please see attach file

Authors did not mention how many replicate they used and if the experiment repeated or not

all figures need to improve 

they used solvents but they did not used it as a control as maybe know the ethanol it self used as sterilize 

English need to revise will 

Author Response

Comments and response to reviewer No. 4

  • Authors did not mention how many replicate they used and if the experiment repeated or not

Response: We agree with the reviewer’s comment. We used three replicates for the antifungal studies from which we performed the statistical analysis. We have updated that information in the Materials and methods section.

  • all figures need to improve

Response: We agree with the reviewer’s comment. We have upgraded the figures as required.

  • they used solvents but they did not used it as a control as maybe know the ethanol it self used as sterilize

Response: We agree with the reviewer’s comment. However, we think there’s a misunderstand of the preparation method of extracts. Ethanolic and water extracts included the usage of both solvents in the preparation only, while the final product was dried to remove any liquified products. In the experimental performance, only milli-Q water was used for the dissolving of both extracts, which was used as a negative control, as well.

  • English need to revise will.

Response: We agree with the reviewer’s comment. We have made excessive lingual, grammatical, and punctuational editing all over the manuscript. 

Comments in attached file:

  • Changes required in the abstract and keywords section.

Response: We agree with the reviewer’s comment. We have upgraded the abstract and keywords section as required.

  • Changes required in the materials and methods section.
  • one night is enough for dry?

Response: We agree with the reviewer’s comment. They were dried for 1-2 days at room temperature (25-30°C), which was enough because of the less humid weather of Riyadh city. We have upgraded the Plant material and preparation of extracts sub-section as required.

  • one ml for one L or one ml for each plate?

Response: We agree with the reviewer’s comment. It’s one ml for each plate. We have upgraded the Antifungal Assay sub-section as required.

  • amount means liquid or you added discs?

Response: We agree with the reviewer’s comment. It’s 6 mm discs from each fungal culture. We have upgraded the Antifungal Assay sub-section as required.

  • what about the solvent?

Response: We agree with the reviewer’s comment. We have clarified that point in the Antifungal Assay sub-section as required.

  • how many replicates you used and are you repeated the experiment?

Response: We agree with the reviewer’s comment. We have clarified that point in the Antifungal Assay sub-section as required.

  • why only ethanolic what about water?

Response: We agree with the reviewer’s comment. However, we thought that there will be no significant changes between both extracts. Also, the limited funding didn’t allow the repetition of that experiment for water extract, as well.

  • what this mean?
  • this growth or inhibition %.

Response: We agree with the reviewer’s comment. We have updated Figure 1 and figure 1 caption as required.

  • is this morphology I do not

Response: We agree with the reviewer’s comment. We have made some rephrasing to clarify these results.

  • where this results?
  • is this your result if yes where this in results?

Response: We agree with the reviewer’s comment. We have made some rephrasing to clarify these points, which belong to other studies. The discussion section was reviewed and updated.

Reviewer 5 Report

Reviewers' comments:

Manuscript Number: microorganisms-2041971

Full Title: The antifungal properties of Tamarix aphylla extract against some plant pathogenic fungi.

Comments: 

The manuscript reported on The antifungal properties of Tamarix aphylla extract against some plant pathogenic fungi. The manuscript needs a detailed editing. It cannot be recommended for publication in the present form. I hope the following points would be helpful for the authors.

- In the Abstract, the authors need to improve with more specific short results and conclusions.

- The Author should add more suitable keywords.

- Introduction is lacking of novelty statement. Please revise and add some recent papers in order to improve the introduction.

- The Materials and Methods section should be detailed especially for the 2.2. Microorganisms, 2.5. Fourier-Transform Infrared Spectroscopy (FTIR), and 2.6. Gas chromatography/mass spectrometry technique (GC/MS). 

- Figures 5 and 6. - Not clear make clear.

- In part SEM: how the energy of the accelerator beam used?

- 3.4. Morphological Characterizations of the Fungal Growth - should be improved.

- Main findings should also be provided in conclusions.

- References: there are recent references in 2021-2022 treating the same subject, you can use.

- Make all references in same format for volume number, page numbers and journal name, because it is difficult to searching and reading.

- Language needs substantial improvement. Please consult a native English speaker or a language editing service.

Based on these, I advise the authors to rectify the above-mentioned errors and we hope to re-evaluate the revised manuscript.

Author Response

Comments and response to reviewer No. 5

  • -In the Abstract, the authors need to improve with more specific short results and conclusions.

Response: We agree with the reviewer’s comment. We have upgraded the abstract as required.

  • The Author should add more suitable keywords.

Response: We agree with the reviewer’s comment. We have upgraded the Keywords as required.

  • Introduction is lacking of novelty statement. Please revise and add some recent papers in order to improve the introduction.

Response: We agree with the reviewer’s comment. We have updated the introduction section with the required information.

  • The Materials and Methods section should be detailed especially for the 2.2. Microorganisms, 2.5. Fourier-Transform Infrared Spectroscopy (FTIR), and 2.6. Gas chromatography/mass spectrometry technique (GC/MS).

Response: We agree with the reviewer’s comment. For the 2.2. Microorganisms sub-section, the species were well-identified by microscope and sequencing in the department plant protection, College of Food and Agricultural Sciences, King Saud University, Riyadh, Saudi Arabia, as been mentioned in the manuscript. For the fungal culture method, it’s described in the Antifungal assay sub-section. We have updated 2.5. Fourier-Transform Infrared Spectroscopy (FTIR), and 2.6. Gas chromatography/mass spectrometry technique (GC/MS) sub-sections with more details, as required.

  • Figures 5 and 6. - Not clear make clear.

Response: We agree with the reviewer’s comment. We have updated the figures as required.

  • In part SEM: how the energy of the accelerator beam used?

Response: We agree with the reviewer’s comment. We have updated the SEM sub-section as required.

  • 4. Morphological Characterizations of the Fungal Growth - should be improved.

Response: We agree with the reviewer’s comment. We have updated sub-section 3.4. as required.

  • Main findings should also be provided in conclusions.

Response: We agree with the reviewer’s comment. We have updated the conclusion part.

  • References: there are recent references in 2021-2022 treating the same subject, you can use.

Response: We agree with the reviewer’s comment. We have added some recent references.

  • Make all references in same format for volume number, page numbers and journal name, because it is difficult to searching and reading.

Response: We agree with the reviewer’s comment. We have updated all references.

  • Language needs substantial improvement. Please consult a native English speaker or a language editing service.

Response: We agree with the reviewer’s comment. We have made excessive lingual, grammatical, and punctuational editing all over the manuscript. 

Round 2

Reviewer 2 Report

The authors considered the suggestions and comments and corrected the manuscript in detail.

Author Response

  • The authors considered the suggestions and comments and corrected the manuscript in detail.

Response: Thanks for the reviewer's comment.

Reviewer 3 Report

After careful reading; the authors have done a great job improving the manuscript and showing the novelty.

One minor correction will enhance the quality for readers:

- The authors need to add marks on SEM and TEM scanning of M. phaseolina photos to clear the effect of these extracts on the morphology and the cell structure.

Author Response

After careful reading; the authors have done a great job improving the manuscript and showing the novelty.

One minor correction will enhance the quality for readers:

- The authors need to add marks on SEM and TEM scanning of M. phaseolina photos to clear the effect of these extracts on the morphology and the cell structure.

Response: Thanks for the reviewer’s comment. We have updated the figures as required.

Reviewer 4 Report

Thanks for you response for the comment 

The Fig still need to be edited For example 

Figure 2. Bar chart for the antifungal properties of effect of T. aphylla water extract. Different concentrations of the extract showed variable inhibitory effects on the growth of different phytopathogenic fungi cultured on PDA agar medium.

it should be  

Effect of different concentration (....................) of T. aphylla water extract on growth of different phytopathogenic fungi. Values in the column followed by different letters indicate significant differences among treatments according to a least significant difference test (p = 0.05). Bars indicate the standard error.

also all fig and Tables like this  

you can see and used the following new references 

DOI: 10.5829/idosi.aejaes.2015.15.4.12571

DOI: 10.9734/JPRI/2022/v34i45B36359

doi.org/10.3390/su142215233

doi.org/10.3390/horticulturae8030197

doi.org/10.17221/14/2011-PPS

doi: 10.31254/phyto.2021.10404

Author Response

Comments and response to reviewer No. 4_round 2

Thanks for your response for the comment

The Fig still needs to be edited. For example

Figure 2. Bar chart for the antifungal properties of effect of T. aphylla water extract. Different concentrations of the extract showed variable inhibitory effects on the growth of different phytopathogenic fungi cultured on PDA agar medium.

it should be  

Effect of different concentration (....................) of T. aphylla water extract on growth of different phytopathogenic fungi. Values in the column followed by different letters indicate significant differences among treatments according to a least significant difference test (p = 0.05). Bars indicate the standard error.

also all fig and Tables like this 

you can see and used the following new references

DOI: 10.5829/idosi.aejaes.2015.15.4.12571

DOI: 10.9734/JPRI/2022/v34i45B36359

doi.org/10.3390/su142215233

doi.org/10.3390/horticulturae8030197

doi.org/10.17221/14/2011-PPS

doi: 10.31254/phyto.2021.10404.

Response: Thanks for the reviewer’s comment. We have updated the figure captions as required.

Reviewer 5 Report

The manuscript can published. The authors have answered the questions.

Author Response

  • The manuscript can published. The authors have answered the questions.

Response: Thanks for the reviewer’s comment.
